

# Relativistic Landau quantization in non-uniform magnetic field and its applications to white dwarfs and quantum information

**Srishty Aggarwal[1*], Banibrata Mukhopadhyay[1†] and Gianluca Gregori[2‡]**

**1** Department of Physics, Indian Institute of Science, Bangalore 560012, India
**2** Department of Physics, University of Oxford, Parks Road, Oxford OX1 3PU, UK

* srishtya@iisc.ac.in, † bm@iisc.ac.in, ‡ gianluca.gregori@physics.ox.ac.uk

## Abstract

We investigate the two-dimensional motion of relativistic cold electrons in the presence of 'strictly' spatially varying magnetic fields satisfying, however, no magnetic monopole condition. We find that the degeneracy of Landau levels, which arises in the case of the constant magnetic field, lifts out when the field is variable and the energy levels of spin-up and spin-down electrons align in an interesting way depending on the nature of change of field. Also, the varying magnetic field splits Landau levels of electrons with zero angular momentum from positive angular momentum, unlike the constant field which only can split the levels between positive and negative angular momenta. Exploring Landau quantization in non-uniform magnetic fields is a unique venture on its own and has interdisciplinary implications in the fields ranging from condensed matter to astrophysics to quantum information. As examples, we show magnetized white dwarfs, with varying magnetic fields, involved simultaneously with Lorentz force and Landau quantization affecting the underlying degenerate electron gas, exhibiting a significant violation of the Chandrasekhar mass-limit; and an increase in quantum speed of electrons in the presence of a spatially growing magnetic field.



# 1   Introduction

The role of magnetic fields in controlling the natural – Earth based to astrophysical – systems from the microscopic to macroscopic scales is well established. From the formation of stars to stellar winds, cosmic rays, accretion disks and jets in X-ray binaries and active galactic nuclei, the magnetic field plays an indispensable role in all the astrophysical systems. In the Earth based systems and laboratory, quantum Hall effect, de Haas Van Alphen effect, vortices, superconductivity, high-resolution NMR and EPR spectroscopies are some of the landmark contributions of high magnetic field physics to the solid state and condensed matter sciences, analytical chemistry and structural biology.

The interaction of strong magnetic field with Fermi gas gives rise to many interesting effects. Two of the main effects are Landau Quantization (hereinafter LQ) [1] and Geometric Phase [2]. Most of the other applications appear to be advanced manifestations of these two. LQ has been well established and discussed in detail for uniform magnetic fields in both non-relativistic [3] as well as relativistic [4] cases. In one hand, non-relativistic Landau effect has been extremely useful in explaining many condensed matter experiments through, e.g., quantum Hall effect, de Haas Van Alphen effect and Shubnikov-de Haas oscillations (see, e.g., [5–8]). On the other hand, relativistic LQ is helpful in resolving many astrophysical mysteries and quantum speed limit of fermions (e.g. [9]). Effect of high magnetic field in neutron stars, particularly in the surface with magnitude $\sim 10^{15}$ G as is proposed in the premise of magnetar based model, is involved with LQ (even if involved with many levels). Further, Das & Mukhopadhyay by taking the stoke of LQ explained the possible existence of super-Chandrasekhar mass white dwarfs [10] and their new mass-limit [11], assuming magnetic fields to be uniform in such white dwarfs. It was also shown that LQ leads to softening the equation of state for neutron stars in the presence of strong magnetic field, though the stiffening effect due to anomalous magnetic moment may overwhelm it [12]. In addition, it was shown by one of the present authors that strong magnetic field induced LQ influences the neutronization threshold and the onset of neutron drip by increasing the density for the former and increasing or decreasing for the latter depending on the magnetic field [13]. It was further confirmed by others [14] showing that the neutron drip line in the crust of highly magnetized star shifts to either higher or lower densities depending on the magnetic field strength. Interestingly, synthetic Landau levels for photons has also been explored [15]. As photons experiencing a Lorentz force develop handedness, they provide opportunities to study quantum Hall physics and topological quantum science.

However, most of the LQ effects mentioned above are probed maintaining the field uniform. In reality, particularly, in the astrophysical cases pointed out above, the magnetic field is never uniform. Note that LQ effects become important only when the gyromagnetic radius is comparable or less than the Compton wavelength of the underlying particles. Moreover, LQ theory based on uniform magnetic field does not suit for non-uniform magnetic fields, if the magnetic field varies in a length scale comparable or shorter than the Compton wavelength of the particles.

Altshuler & Ioffe [16] were the first to discuss the motion of fast particles in the strongly fluctuating magnetic field and showed analytically how fluctuations result in phase incoherence. Following their work, others (see, e.g., [17,18]) discussed multiple aspects of the motion of particles in magnetic fields, taking into considerations strong as well as weak but random fluctuations over uniform field, spatially modulated magnetic fields lifting degeneracy in the nonrelativistic regime, etc. There are other explorations of the effects of random magnetic fields to the electron gas and LQ, and their implications to, e.g., composite fermions and the quantum critical point [19,20].

What if, the field is completely non-uniform like in astrophysical systems and plasma? In white dwarfs, neutron stars, as well as main sequence stars, e.g. in Sun, it is almost certain that field varies from the center to surface. If the magnetar is a highly magnetized neutron star, while its surface field is observationally inferred to be $\sim 10^{15}$ G, its central field could be orders of magnitude higher. Similarly, highly magnetized white dwarfs (B-WDs) violating Chandrasekhar-limit significantly with a new mass-limit [11] are argued to have central and surface fields $\geq 10^{15}$ G and $\leq 10^{12}$ G respectively [21–24], hence LQ clearly can not be avoided therein. Note that at high densities in B-WDs, the Coulomb interactions turn out to be negligible compared to Fermi energy [10]. Therefore electric fields are negligible compared to the magnetic fields therein, particularly if the field is not time varying. Hence, one can ignore the QED induced effect of pair creation in B-WDs [10,25,26], which can not happen in magnetic fields alone. Indeed, field magnitude decaying with density was proposed earlier for neutron stars and white dwarfs [27,28] in an ad-hoc basis and predicted its influence on the mass, radius, and luminosity (see, e.g., [24], for a latest application). However, there was no account for the associated potential and Maxwell's equations in such variation of magnetic field. Hence the effect of magnetic field was considered in an adiabatic approximation and hence LQ in uniform field. However, such systems with magnetic field having variation over spatial region as an explicit function of distance has not gained direct attention till date.

We aim here to explore the change in LQ effect on the energy levels of relativistic electrons in the presence of mainly a decaying magnetic field, but also a growing magnetic field. The chosen field profiles, as demonstrated below, are in accordance with Maxwell's equations of electromagnetism. Since LQ is a quantum phenomenon, it is expected to be affected only if the variation in magnetic field takes place at the quantum scale. Therefore, we probe the variation of magnetic field at scales of the order of gyromagnetic radius which is less than or of the order of Compton wavelength of electron, determined by the chosen magnetic fields.

It is generally expected that the magnetic field and density vary in a star as decreasing functions of its radial coordinate. Therefore, the magnetic field is expected to be varying with the density. As the density is expected to be highest at the center and lowest at the surface, it is a reasonable assumption that the magnetic field too follows the same trend, as proposed earlier [27]. Such a field profile has been extensively used for neutron stars and white dwarfs with appropriate parameters (see, e.g., [24,29]), which may induce a sharp variation of the field in a short spatial scale, depending upon the parameters.

As an application of non-uniform and growing magnetic field, we also show its role in attaining higher quantum speed, i.e. transition speed from one energy level to other, of electrons as compared to uniform magnetic field. This could help in achieving faster processing speed

of quantum computers along with other applications.

For relativistic electrons, the splitting of levels due to spin has significant contribution in determining the energy spacing and overall structure of energy levels and, hence, cannot be treated perturbatively. For uniform magnetic field, it, in fact, leads to doubly degenerate levels [3]. We show and investigate, how the degeneracy due to spin arising in constant magnetic field breaks down when the field is variable.

In the next section (Sec.2), we establish the formalism of the problem of variable magnetic field. Subsequently, we review the solution for uniform magnetic field in Sec.2.1 and explore the effect of the non-uniform magnetic field in detail including the effective potential observed by the electron in Sec.2.2. The computational methods to determine the eigenvalue spectrum are also outlined in Sec.2.2. The solutions of established equations are shown in Sec.3.1 and Sec.3.2. The underlying thermodynamics and equation of state (EoS) are explored in Sec.4 and its implications, in an astrophysical context and a quantum information, have been enlightened in Sec.5. We conclude in Sec.6 by highlighting the key points of this work and its various implications.

## 2 Dirac equation for electrons and its solution in the presence of magnetic fields

For electron of mass $m_e$ and charge $q$ $(-e)$, the Dirac equation in the presence of magnetic field is given by

$$i\hbar\frac{\partial\Psi}{\partial t} = \left[c\boldsymbol{\alpha}\cdot\left(-i\hbar\boldsymbol{\nabla}-\frac{q\mathbf{A}}{c}\right)+\beta m_e c^2\right]\Psi, \tag{1}$$

where $\boldsymbol{\alpha}$ and $\beta$ are Dirac matrices, $\mathbf{A}$ is the vector potential, $\hbar = h/2\pi$ with $h$ being Planck's constant and $c$ is the speed of light. For stationary states, we can write

$$\Psi = e^{-i\frac{Et}{\hbar}}\begin{bmatrix}\chi\\\phi\end{bmatrix}, \tag{2}$$

where $\Phi$ and $\chi$ are 2-component objects/spinors. We consider the Pauli-Dirac representation in which

$$\alpha = \begin{bmatrix}0 & \boldsymbol{\sigma}\\\boldsymbol{\sigma} & 0\end{bmatrix}, \qquad \beta = \begin{bmatrix}I & 0\\0 & -I\end{bmatrix}, \tag{3}$$

where each block represents a $2 \times 2$ matrix and $\boldsymbol{\sigma}$ represents three components of the Pauli matrices together in a vector. Hence Eq. (1) reduces to

$$(E-m_e c^2)\chi = c\boldsymbol{\sigma}\cdot\left(-i\hbar\boldsymbol{\nabla}-\frac{q\mathbf{A}}{c}\right)\phi, \tag{4}$$

$$(E+m_e c^2)\phi = c\boldsymbol{\sigma}\cdot\left(-i\hbar\boldsymbol{\nabla}-\frac{q\mathbf{A}}{c}\right)\chi. \tag{5}$$

Decoupling them for $\chi$, we obtain

$$(E^2-m_e^2 c^4)\chi = \left[c\boldsymbol{\sigma}\cdot\left(-i\hbar\boldsymbol{\nabla}-\frac{q\mathbf{A}}{c}\right)\right]^2\chi. \tag{6}$$

Defining $\boldsymbol{\pi} = -i\hbar\boldsymbol{\nabla}-q\mathbf{A}/c$ and using the identity $(\boldsymbol{\sigma}\cdot\boldsymbol{\pi})(\boldsymbol{\sigma}\cdot\boldsymbol{\pi}) = \pi^2 - q\hbar\boldsymbol{\sigma}\cdot\boldsymbol{B}/c$, Eq. (6) reduces to

$$(E^2-m_e^2 c^4)\chi = \left[c^2\left(\pi^2-\frac{q\hbar}{c}\boldsymbol{\sigma}\cdot\mathbf{B}\right)\right]\chi, \tag{7}$$

such that the antiparticle wavefunction $\phi = -\chi$ [4]. We solve Eq. (7) for a variable magnetic field in cylindrical coordinates. As there is no fixed law for the variation of magnetic field in nature, except that it should satisfy Maxwell's equations, we choose a simple power law variation of the magnetic field, given by

$$\mathbf{B} = B_0 \rho^n \hat{z}, \tag{8}$$

in cylindrical coordinates $(\rho, \phi, z)$. Such a field profile satisfies no monopole condition $(\nabla \cdot \mathbf{B} = 0)$ and according to Ampére's law produces current. See appendix for total Lagrangian and from Lagrangian equation of motion how to obtain the Dirac and Maxwell's equations. For the present purpose of underlying quantum physics, our interest is in the Dirac equation. However, in certain applications, e.g. in stellar physics, the underlying Maxwell's equation needs to be paid attention in order to include classical Lorentz force. The chosen magnetic field profile also assures the decaying nature of the field away from the source if $n < 0$ which is a common feature, particularly in stellar physics. Also, the same profile with $n > 0$ can be applicable for a system with spatially growing field satisfying other physics intact. Using a gauge freedom for the vector potential $\mathbf{A}$, we choose

$$\mathbf{A} = B_0 \frac{\rho^{n+1}}{n+2} \hat{\phi} = A\hat{\phi}. \tag{9}$$

Hence,

$$\pi^2 \chi = \left[ \hat{p}_\rho^2 + \left( \hat{p}_\phi - \frac{qA}{c} \right)^2 + \hat{p}_z^2 \right] \chi, \tag{10}$$

where $\hat{p}_{\rho,\phi,z}$ denote operators. Noticing that $\phi$ and $z$ are ignorable coordinates, the solution of Eq. (7) can be written as

$$\chi = e^{i\left(m\phi + \frac{p_z}{\hbar}z\right)} R(\rho), \tag{11}$$

where $R(\rho)$ is a two-component matrix, '$m\hbar$' is the angular momentum of the electron and $p_z$ is the eigenvalue of momentum in the $z-$direction. Therefore, Eq. (10) becomes

$$\pi^2 R = -\hbar^2 \left[ \frac{\partial^2}{\partial \rho^2} + \frac{1}{\rho} \frac{\partial}{\partial \rho} - \frac{m^2}{\rho^2} \right] R(\rho) + \left[ \frac{q^2 A^2}{c^2} + \frac{2q\hbar m A}{c\rho} + p_z^2 \right] R(\rho). \tag{12}$$

From Eqs. (7), (10) and (12) and substituting $q = -e$, we obtain

$$\left( \frac{E^2 - m_e^2 c^4}{c^2} - p_z^2 \right) R(\rho) = -\hbar^2 \left[ \frac{\partial^2}{\partial \rho^2} + \frac{1}{\rho} \frac{\partial}{\partial \rho} - \frac{m^2}{\rho^2} \right] R(\rho)$$

$$+ \left[ \frac{e^2 A^2}{c^2} - \frac{2e\hbar m A}{c\rho} + \frac{e\hbar}{c} (\sigma_z B) \right] R(\rho). \tag{13}$$

There will be two independent solutions for $R(\rho)$, which can be taken, without loss of generality, to be the eigenstates of $\sigma_z$, with eigenvalues $\pm 1$. Thus if we choose two independent solutions of the form

$$R_+(\rho) = \begin{bmatrix} \tilde{R}_+(\rho) \\ 0 \end{bmatrix}, \qquad R_-(\rho) = \begin{bmatrix} 0 \\ \tilde{R}_-(\rho) \end{bmatrix},$$

such that $\sigma_z R_\pm = \pm R_\pm$, Eq. (13) becomes

$$\tilde{P}\tilde{R}_\pm = -\hbar^2 \left[ \frac{\partial^2}{\partial \rho^2} + \frac{1}{\rho} \frac{\partial}{\partial \rho} - \frac{m^2}{\rho^2} \right] \tilde{R}_\pm + \left[ \frac{e^2 A^2}{c^2} - \frac{2e\hbar m A}{c\rho} \pm \frac{e\hbar}{c} B \right] \tilde{R}_\pm, \tag{14}$$

where

$$\tilde{P} = \left( \frac{E^2 - m_e^2 c^4}{c^2} - p_z^2 \right). \tag{15}$$

Dividing Eq. (14) by $m_e^2 c^2$, we have an eigenvalue equation as

$$\alpha \tilde{R}_{\pm} = -\left( \frac{\hbar}{m_e c} \right)^2 \left[ \frac{\partial^2}{\partial \rho^2} + \frac{1}{\rho} \frac{\partial}{\partial \rho} - \frac{m^2}{\rho^2} \right] \tilde{R}_{\pm} + \left[ \frac{e^2 A^2}{m_e^2 c^4} + \frac{e\hbar}{m_e^2 c^3} \left( -\frac{2mA}{\rho} \pm B \right) \right] \tilde{R}_{\pm} \tag{16}$$

$$= -\lambda_e^2 \left[ \frac{\partial^2}{\partial \rho^2} + \frac{1}{\rho} \frac{\partial}{\partial \rho} - \frac{m^2}{\rho^2} \right] \tilde{R}_{\pm} + \left[ \left( \frac{kB_0 \rho^{n+1}}{n+2} \right)^2 + k\lambda_e \left( -\frac{2m}{n+2} \pm 1 \right) B_0 \rho^n \right] \tilde{R}_{\pm}, \tag{17}$$

where $\alpha = \frac{\tilde{P}}{m_e^2 c^2} = (\epsilon^2 - 1 - x_z^2)$ which is, in fact, square of dimensionless momentum and acting as an eigenvalue of the problem, $\epsilon = \frac{E}{m_e c^2}$ (dimensionless energy), $x_z = \frac{p_z}{m_e c}$ (dimensionless momentum along $z-$direction), $\lambda_e = \frac{\hbar}{m_e c}$ (Compton wavelength of electrons), $k = \frac{e}{m_e c^2}$. Note that this $\alpha$ should not be confused with Dirac $\boldsymbol{\alpha}$ matrix.

## 2.1 Uniform Magnetic Field ($n = 0$)

For constant magnetic field, Eq. (17) becomes

$$\alpha \tilde{R}_{\pm} = -\lambda_e^2 \left[ \frac{\partial^2}{\partial \rho^2} + \frac{1}{\rho} \frac{\partial}{\partial \rho} - \frac{m^2}{\rho^2} \right] \tilde{R}_{\pm} + \left[ \left( \frac{kB_0 \rho}{2} \right)^2 + k\lambda_e (-m \pm 1) B_0 \right] \tilde{R}_{\pm}. \tag{18}$$

The above equation can be solved analytically similar to its non-relativistic counterpart [3]. Now defining $\xi = \left( \frac{kB_0}{2\lambda_e} \right) \rho^2$, Eq. (18) can be written as

$$\xi \tilde{R}''_{\pm} + \tilde{R}'_{\pm} + \left( -\frac{1}{4}\xi + \beta_{\mp} - \frac{m^2}{4\xi} \right) \tilde{R}_{\pm} = 0, \tag{19}$$

where

$$\beta_{\mp} = \frac{\alpha}{2\lambda_e k B_0} + \left( \frac{m}{2} \mp \frac{1}{2} \right)$$

and double-prime ($''$) and prime ($'$) respectively denote double and single derivatives with respect to $\rho$. At $\xi \to \infty$, the solution of Eq. (19) gives as $\tilde{R}_{\pm} \sim e^{-\frac{\xi}{2}}$, and for $\xi \to 0$ as $\tilde{R}_{\pm} \sim \xi^{\frac{|m|}{2}}$. Accordingly, we seek a solution of the form

$$\tilde{R}_{\pm} = e^{-\frac{\xi}{2}} \xi^{\frac{|m|}{2}} w(\xi). \tag{20}$$

Thence equation for $w(\xi)$ satisfies the confluent hypergeometric function so that

$$w = F\left[ -\left( \beta_{\mp} - \frac{|m|}{2} - \frac{1}{2} \right), |m| + 1, \xi \right]. \tag{21}$$

For the wavefunction to be finite everywhere, the quantity $\left( \beta_{\mp} - \frac{|m|}{2} - \frac{1}{2} \right)$ must be a non-negative integer $\nu$. Hence, the values of $\alpha$ are given by

$$\alpha_{\nu} = 2k\lambda_e B_0 \left( \nu + \frac{|m|}{2} - \frac{m}{2} + \frac{1}{2} \pm \frac{1}{2} \right), \tag{22}$$

where $m$ is the azimuthal quantum number. One can easily see from Eq. (22) that ground state energy (corresponding to $\alpha_0$) is 0 and all the other energy levels are doubly degenerate. Also, energies are same for $m = 0$ and $> 0$. Finally $\tilde{P}$ in Eq. (15) turns out to be $2\nu B_0/B_c$, where $B_c = m_e^2 c^3/e\hbar$, the Schwinger limit of pair production, so that

$$E^2 = p_z^2 c^2 + m_e^2 c^4 \left( 1 + 2\nu \frac{B_0}{B_c} \right). \tag{23}$$

## 2.2 Non-Uniform Magnetic Field ($n \neq 0$)

We know that analytic solutions exist for some special potentials only, which include harmonic oscillator, hydrogen-atom and Morse-oscillator. For the presently chosen potential, however, we are not able to find solutions analytically. Therefore, we use computational methods to find eigenvalues $\alpha_\nu$ at different levels $\nu$ for different $n$. Let us first explore the asymptotic behaviour of $\tilde{R}_\pm$ (the asymptotic behaviour is same for $\tilde{R}_+$ and $\tilde{R}_-$).

As $\rho \to 0$, Eq. (17) becomes

$$-\lambda_e^2 \left[ \frac{\partial^2}{\partial \rho^2} + \frac{1}{\rho} \frac{\partial}{\partial \rho} \right] \tilde{R}_\pm = 0 \,. \tag{24}$$

Hence, $\tilde{R}_\pm \to C_1 + C_2 \log(\rho)$, $C_1$ and $C_2$ being constants. Since $\log(\rho)$ blows up at $\rho \to 0$, to seek for a finite solution throughout, we set $C_2 = 0$. Hence, as $\rho \to 0$

$$\tilde{R}_\pm \to C_1, \quad \tilde{R}'_\pm \to 0 \,. \tag{25}$$

For $\rho \to \infty$, however, for $n \leq 0$, Eq. (17) turns out to be

$$\left[ -\lambda_e^2 \frac{\partial^2}{\partial \rho^2} + \left( \frac{kB_0 \rho^{n+1}}{n+2} \right)^2 \right] \tilde{R}_\pm = 0 \,. \tag{26}$$

Thus,

$$\tilde{R}_\pm \to e^{-\left[ \frac{kB_0}{\lambda_e(n+2)} \right] \frac{\rho^{n+2}}{n+2}} \quad \text{as } \rho \to \infty \,. \tag{27}$$

There are many different methods to solve Eq. (17) including 'Finite Difference' method and 'Shooting and Matching' method. We obtain most accurate solutions with the 'Shooting and Matching' method [30], where the relative error between results in exact theory and computation is below 0.0004 for lower Landau levels and it never exceeds 0.002 even in higher

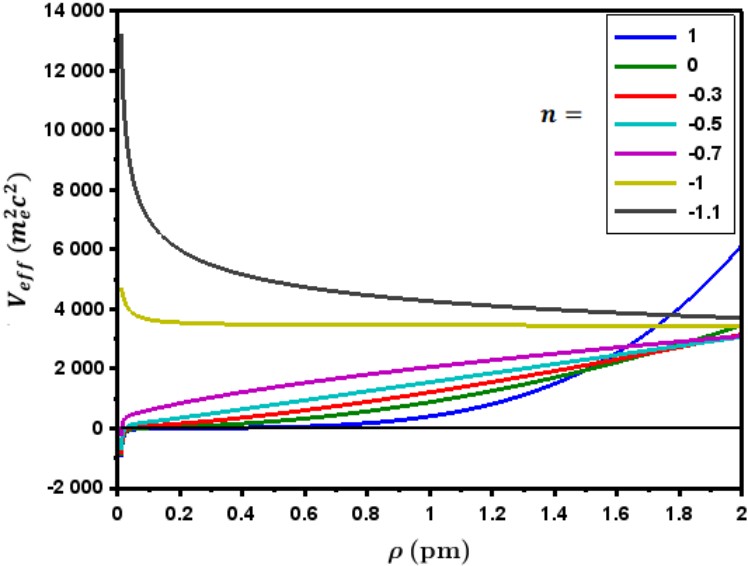

Figure 1: The variation of effective potential for different $n$ for $B_0 = 10^{15}$ G pm$^{-n}$. Here, the black horizontal line represents $V_{eff} = 0$. Various potentials at $\rho = 1$ pm from bottom to top successively are for $n = 1, 0, -0.3, -0.5, -0.7, -1, -1.1$.

Table 1: Comparison of the eigenvalues obtained from numerical computation (column two) and theory (column three) for constant magnetic fields ($n = 0$) with $B_0 = 10^{15}$ G pm$^{-n}$.

| $\nu$ | $\alpha_{comp}$ | $\alpha_{th}$ | relative error |
|---|---|---|---|
| 0 | 22.200623 | 22.2094 | 0.0003952 |
| 1 | 66.616364 | 66.6282 | 0.0001776 |
| 2 | 111.03531 | 111.047 | 0.0001053 |
| 3 | 155.4541 | 155.4658 | 0.0000752 |
| 4 | 199.87289 | 199.8846 | 0.0000586 |
| 5 | 244.29169 | 244.3034 | 0.0000479 |
| 6 | 288.7104 | 288.7222 | 0.0000409 |
| 7 | 333.12916 | 333.141 | 0.0000355 |
| 8 | 377.54795 | 377.5598 | 0.0000314 |
| 9 | 421.96673 | 421.9786 | 0.0000281 |

levels for the constant field case (see Table 1). The differential equation is solved using "ode rk" command in scilab[1] which is traditional adaptive Runge-Kutta method. In order to obtain the initial conditions, we use our knowledge for the behavior of $\tilde{R}_{\pm}$ at $\rho \to 0$ and then set $C_1 = 1$. Thus, we have initial conditions as $\tilde{R}_{\pm}(\rho \to 0) = 1$, $\tilde{R}'_{\pm}(\rho \to 0) = 0$. Ideally, initial conditions should be defined at $\rho = 0$, but many terms in Eq. (18) blows up at $\rho = 0$. Hence, we define the initial conditions at $\rho = 10^{-10}$ picometer (pm) which is equivalent to 0 compared to even the minimum gyromagnetic radius for our field of interest, which is of the order of pm. We express magnetic fields in units of G and length in pm for solving Eq. (17). Thus, $B_0 = |\mathbf{B}| = B$ at 1 pm.

Also, to remove the diverging nature of magnetic field near the origin with $n < 0$, we choose

$$B = B_0 (\rho + \rho_0)^n, \tag{28}$$

where $\rho_0$ could be chosen to be a very small number as compared to the scale of wavefunction. We choose it to be equal to $10^{-5}$ pm. As long as $\rho_0$ is very small, this choice does not effect the solutions.

To determine the effective potential experienced by electrons, let $\tilde{R}_{\pm}(\rho) = \frac{u_{\pm}(\rho)}{\sqrt{\rho}}$. Then, Eq. (17) becomes

$$\alpha u_{\pm} = \left(-\lambda_e^2 \frac{\partial^2}{\partial \rho^2} + V_{eff}\right) u_{\pm}, \tag{29}$$

where

$$V_{eff} = -\lambda_e^2 \left[\frac{1}{4\rho^2} - \frac{m^2}{\rho^2}\right] + \left(\frac{kB_0\rho^{n+1}}{n+2}\right)^2 + k\lambda_e\left(-\frac{2m}{n+2} \pm 1\right)B_0\rho^n.$$

We show the variation of $V_{eff}$ for different $n$ in Figure 1. It is seen that for $n \leq -1$, potential is completely repulsive whose solution will depend on the distance from the source (origin of the system) upto which a particle can move. Therefore, the energy eigenvalues for such cases depend upon where we put a hard wall making the system equivalent to confining the electron in a box. However we do not want to apply any such restrictions on the electron. Moreover, this nature of variation is not realistic, particularly in astrophysical scenarios. We therefore restrict our analysis to cases for $n > -1$.

---

[1]see, https://help.scilab.org/docs/6.0.0/en_US/ode.html.

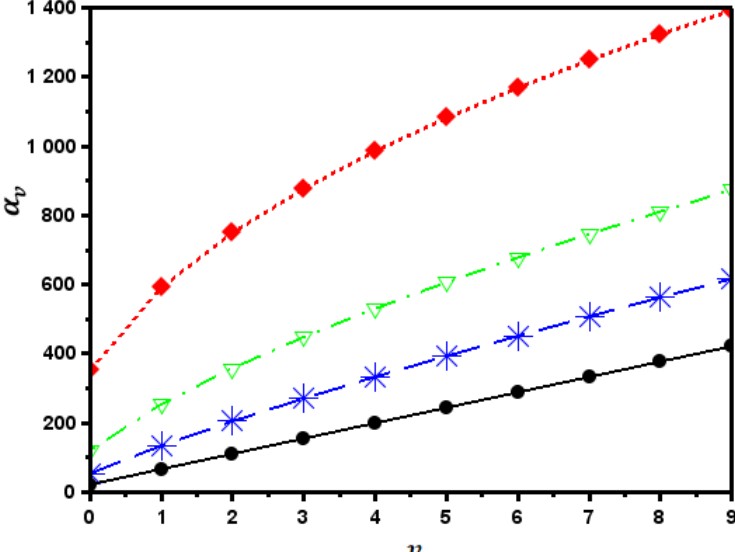

Figure 2: The variation of eigenvalue with eigen-index for $B_0 = 10^{15}$ G pm$^{-n}$, when $n = 0$ (black solid circles), -0.3 (blue dashed asterisks), -0.5 (green dot-dashed triangles) and -0.7 (red dotted diamonds). The lines represent the curves fitted with constants of Eq. (31).

## 3 Dispersion relations

### 3.1 Excluding Zeeman effect

First, we investigate the effect of variation of magnetic fields on the energy levels $\alpha_v$ excluding Zeeman splitting for $m = 0$. Thus, Eq. (17) becomes

$$\alpha'_v \tilde{R} = -\lambda_e^2 \left[ \frac{\partial^2}{\partial \rho^2} + \frac{1}{\rho} \frac{\partial}{\partial \rho} \right] \tilde{R} + \left( \frac{kB_0 \rho^{n+1}}{n+2} \right)^2 \tilde{R}, \tag{30}$$

where $\alpha'_v$ is the energy level excluding Zeeman effect. Figure 2 shows how the spacing of energy levels modifies for different $n : -1 < n \leq 0$. As seen in the figure, as $n$ decreases, energy levels rise up. It is seen from Figure 1 that with the decrease of $n$ (but for $> -1$), potential with increasing $\rho$ crosses the 0 at a smaller distance and thereby becoming repulsive closer to the origin, compared to that of a larger $n$. We know that a particle is more stable if it is in an attractive potential regime and has lower energy. If a particle feels a repulsive potential, it requires more energy to stay in that region, thus explaining the behavior of eigenvalues seen in Figure 2. In simpler words, this increase in energy eigenvalues can be understood as follows. These variations of eigenvalues are for a fixed $B_0$, which is $B$ at 1 pm, when $B$ keeps increasing to a much higher value near the source for lower $n$, thereby, increasing the average magnetic field and, hence, raising the energy levels.

Also, with the decrease of $n$, dispersion energies become highly non-linear, i.e. the difference between two successive levels, while initially is very large, then decreases much faster for smaller $n$, which is physically related to the chosen profile of magnetic field. Due to the faster decaying nature of field, electrons observe a very strong magnetic field near the center, thereby, having significant discretion of energies, for a fixed $B_0$. As it moves little away from the center, magnetic field weakens and, hence, the spacing of levels decreases. One can expect

Table 2: The values of the constants of Eq. (31) for various $n$. Here $B_0$ in Eq. (31) is chosen in the units of $10^{15}$ G pm$^{-n}$ to obtain $C_2$.

| $n$ | $C_3$ | $C_4$ | $C_5$ | $C_6$ |
|---|---|---|---|---|
| 0 | 44.4188 | 1 | 0.5 | 1 |
| -0.1 | 56 | 1.0519 | 0.50 | 0.9467 |
| -0.2 | 72.5 | 1.111 | 0.4934 | 0.8878 |
| -0.3 | 97 | 1.18 | 0.488 | 0.8224 |
| -0.4 | 134.63 | 1.25 | 0.486 | 0.749 |
| -0.5 | 195.66 | 1.33 | 0.484 | 0.665 |
| -0.6 | 301 | 1.43 | 0.482 | 0.5702 |
| -0.7 | 494 | 1.54 | 0.476 | 0.4609 |
| -0.8 | 878.9 | 1.667 | 0.475 | 0.33 |
| -0.9 | 1706 | 1.818 | 0.48 | 0.191 |

a larger change in the energy level gaps, if field decays more rapidly, what is seen in Figure 2.

Since an analytical solution is not easy to obtain, we try to figure out the possible expression for the energy dispersion relation using a suitable ansatz and data fitting. Based on the analogy of constant field case, we suggest the ansatz of the form

$$\alpha'_\nu = C_3 B_0^{C_4} (\nu + C_5)^{C_6}, \tag{31}$$

where $C_3$, $C_4$, $C_5$ and $C_6$ are constants whose values depend on $n$. Table 2 shows the values of these constants that we obtain by fitting numerical data for lower levels when $-0.9 \le n \le 0$. It is interesting to note that

$$C_4 + C_6 = 2 \tag{32}$$

and

$$C_4(n) = \frac{2}{n+2}. \tag{33}$$

## 3.2 Including Zeeman Effect

Now let us obtain the eigenvalues for the entire Eq. (17). Figure 3 shows the eigenvalues for $m = 0$ with (a) $B_0 = 10^{15}$ G pm$^{-n}$, and (b) $B_0 = 5 \times 10^{14}$ G pm$^{-n}$, where different markers distinctly indicate the levels for $-\boldsymbol{\sigma} \cdot \boldsymbol{B}$ ($-B_0$) and $+\boldsymbol{\sigma} \cdot \boldsymbol{B}$ ($+B_0$). To give a better idea about the variation of eigenvalues and the splitting of levels, we fix $\epsilon$ to the Fermi energy $\epsilon_F = 20$ and then obtain $B_0$ and corresponding eigenvalues for one-level, two-level and three-level systems, enlisted in Table 3. There are many interesting results what can be inferred from Figure 3 and Table 3.

The levels which are doubly degenerate in the presence of a constant magnetic field turn out to be non-degenerate when the field varies. A diagram corresponding to the solution of Eq. (17) for the splitting energy and lifting degeneracy with varying field as compared to the constant field case is shown in Figure 4. The trend of splitting is really a nice site for observation. The energy level corresponding to $+\boldsymbol{\sigma} \cdot \boldsymbol{B}$ of ground level, which overlaps with the energy level corresponding to $-\boldsymbol{\sigma} \cdot \boldsymbol{B}$ of first excited level for $n = 0$, becomes a little higher than the energy level for $-\boldsymbol{\sigma} \cdot \boldsymbol{B}$ of first excited level for $n = -0.3$, and lies nearly in the middle of $-\boldsymbol{\sigma} \cdot \boldsymbol{B}$ of first and second excited energy levels for $n = -0.5$. This further falls in closer to the energy level for $-\boldsymbol{\sigma} \cdot \boldsymbol{B}$ of second excited state for $n = -0.7$. In fact, for $n = -0.9$ the eigenvalue for the $+\boldsymbol{\sigma} \cdot \boldsymbol{B}$ of ground level is even larger than the $-\boldsymbol{\sigma} \cdot \boldsymbol{B}$ of third excited level, as shown in Figure 5.

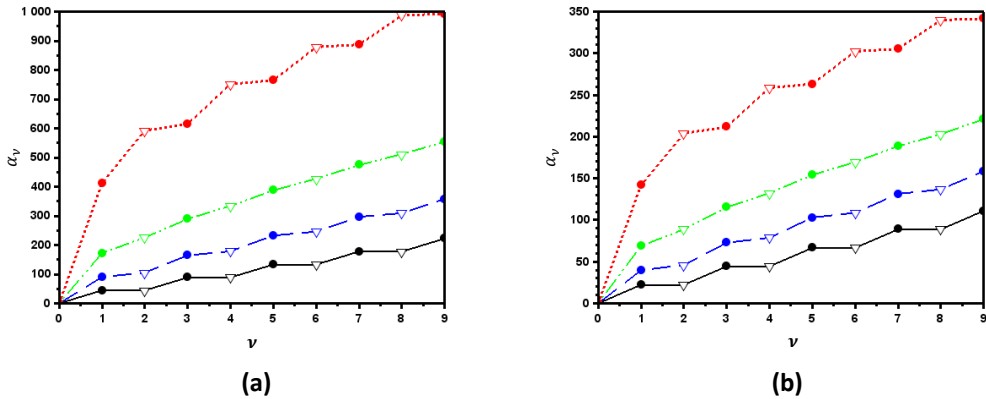

Figure 3: The variation of eigenvalue with the eigen-index for $n = 0$ (black solid line), -0.3 (blue dashed line), -0.5 (green dot-dashed line) and -0.7 (red dotted line) for (a) $B_0 = 10^{15}$ G pm$^{-n}$, and (b) $B_0 = 5 \times 10^{14}$ G pm$^{-n}$, and $m = 0$. Here the levels for $-\boldsymbol{\sigma} \cdot \boldsymbol{B}$ ($-B_0$) and $+\boldsymbol{\sigma} \cdot \boldsymbol{B}$ ($+B_0$) are marked by the solid circles and triangles respectively.

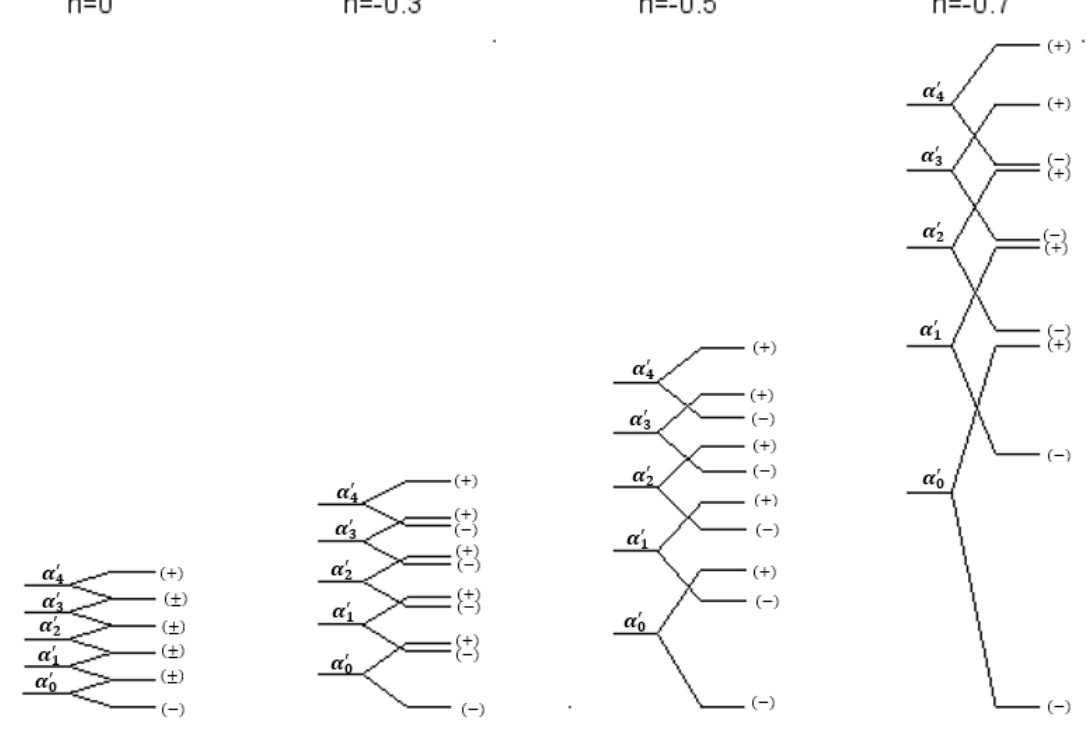

Figure 4: Schematic diagram showing splitting of energy levels with constant and varying magnetic fields for $n = 0, -0.3, -0.5, -0.7$.

Table 3: The variation of $B_0$ for different $n$ for one-level, two-level and three-level systems at $\epsilon = 20$. For $n = 0$, all the levels are doubly degenerate. For $n \neq 0$ all eigenvalues are different independent of energy levels and splits $+\boldsymbol{\sigma} \cdot \boldsymbol{B}$ and $-\boldsymbol{\sigma}.\boldsymbol{B}$. Here '(+)' denotes $\alpha_\nu$ with $+\boldsymbol{\sigma} \cdot \boldsymbol{B}$ and the rest is for $-\boldsymbol{\sigma} \cdot \boldsymbol{B}$.

| $n$ | $\nu_m$ | $B_0$ ($10^{15}$ G pm$^{-n}$) | $\alpha_0$ | $\alpha_1$ | $\alpha_2$ | $\alpha_3$ |
|---|---|---|---|---|---|---|
| 0 | 1 | 8.98 | 0.00976 | 398.866 | 797.397 | 1196.62 |
| | 2 | 4.49 | 0.0152 | 199.461 | 398.879 | 598.311 |
| | 3 | 2.994 | 0.00976 | 133.0327 | 266.0217 | 398.98 |
| -0.3 | 1 | 3.546 | 0.0468 | 399.237 | (+)468.26 | 729.245 |
| | 2 | 3.095 | 0.00312 | 340.2 | (+)399.06 | 621.398 |
| | 3 | 2.125 | 0.002 | 218.28 | (+)256.37 | 399.255 |
| -0.5 | 1 | 1.876 | 0.0428 | 399.15 | (+)532.63 | 669.2 |
| | 2 | 1.533 | 0.006 | 304.718 | (+)399.19 | 511.24 |
| | 3 | 1.275 | 0.0498 | 238.348 | (+)312.29 | 399.897 |
| -0.7 | 1 | 0.979 | 0.000 | 399.44 | (+)573.177 | 595.61 |
| | 2 | 0.744 | 0.00217 | 278.27 | (+)399.3 | 413.934 |
| | 3 | 0.755 | 0.0003 | 267.83 | (+)384.34 | 399.345 |

Note that, ground level always lies at 0 for all $n$. Thus, the physical effects arisen due to the electrons being in ground level only will remain unaltered if a constant field is replaced by a variable field or if there are little inhomogeneities within the constant field background. However, other phenomena that involve with multiple Landau levels are ought to get modified due to unequal spacing and change of degeneracy of levels in non-uniform fields.

Figure 6 shows a sample set of wavefunctions in first few levels. It is clear that wavefunctions fully decay in the region used to determine the eigenvalues. This ensures the correctness of eigenvalues obtained from our computation.

### 3.2.1 Dispersion relation and non-linearity

In order to obtain the dispersion relation, we propose the ansatz for the shift of eigenvalues from the previous case, given by

$$\alpha_\nu = \alpha'_\nu \pm D_1 \, B_0^{D_2} \, (\nu + D_3)^{D_4} \,, \tag{34}$$

where $D_1, D_2, D_3$ and $D_4$ are constants. With many trials and tribulations, we are able to obtain these constants till $n = -0.5$. However, the eigenvalues of very low levels (ground to third) for $+\boldsymbol{\sigma} \cdot \boldsymbol{B}$ do not satisfy these relations exactly, which show that the effect of $\pm\boldsymbol{\sigma} \cdot \boldsymbol{B}$ is not equal near the origin. This confirms that the effect of change in potential on the particle is non-linear and hence supports the power-law ansatz of our proposed dispersion relation. To make it lucid, there is an equal change in the potential due to $-\boldsymbol{\sigma} \cdot \boldsymbol{B}$ and $+\boldsymbol{\sigma} \cdot \boldsymbol{B}$, but when we compute the differences for the same with respect to $\alpha'_\nu$, they follow slightly different trends, which imply that the equal decrease and increase in potential does not have same effect proving the net non-linear dispersion relation for variable magnetic field.

We try to refine the constants in Eq. (34) by assuming that they must have some particular relation with the constants of Eq. (31). The relations for $n \geq -0.5$ come out to be

$$D_1 = C_3 \times C_5; \quad D_2 = C_4; \quad D_3 = C_5; \quad D_4 = C_6 - 1. \tag{35}$$

As $n$ lowers further below $-0.5$, the non-linearity in potential increases so much that the eigenvalues show large deviation from these relations till higher levels ($\nu_m = 10-50$). Hence,

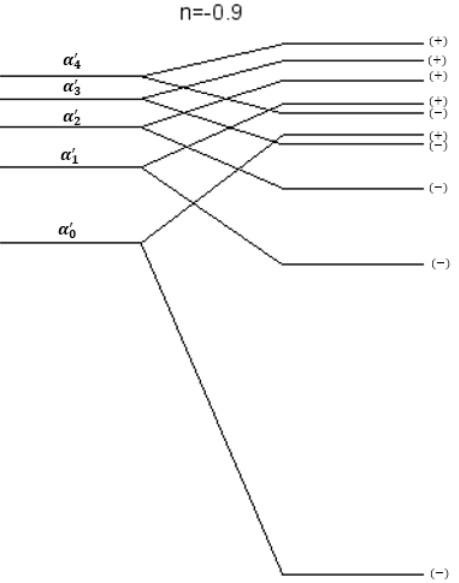

Figure 5: Schematic diagram showing splitting of energy levels with $n = -0.9$.

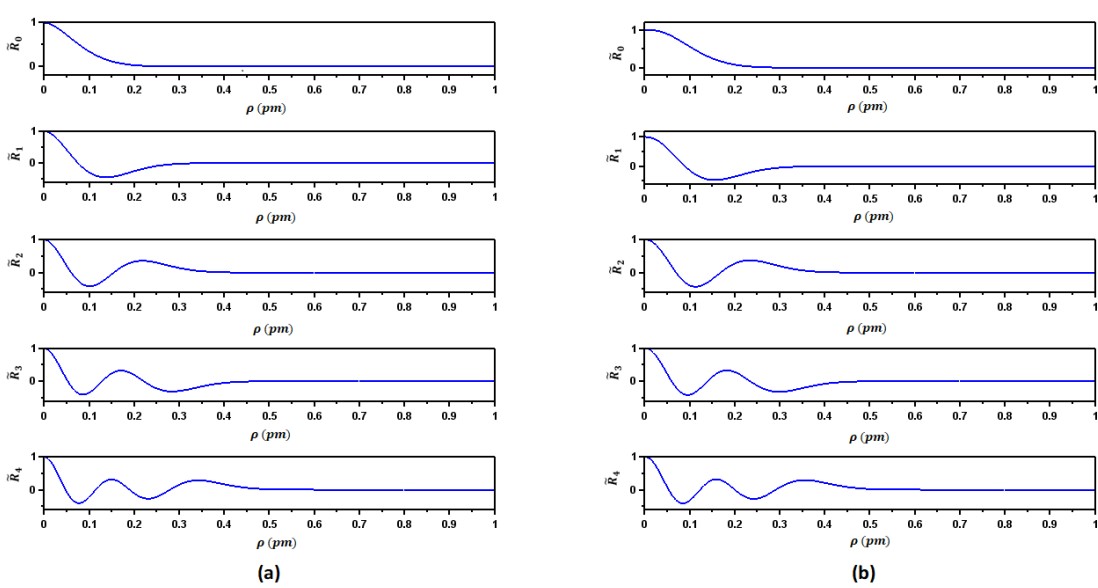

Figure 6: Wavefunctions for $B_0 = 10^{15}$ G pm$^{-n}$, $n = -0.3$ and $m = 0$ for $\nu = 0$ to 4 from the top to bottom panels respectively, for (a) $-\sigma \cdot \mathbf{B}$, and (b) $+\sigma \cdot \mathbf{B}$.

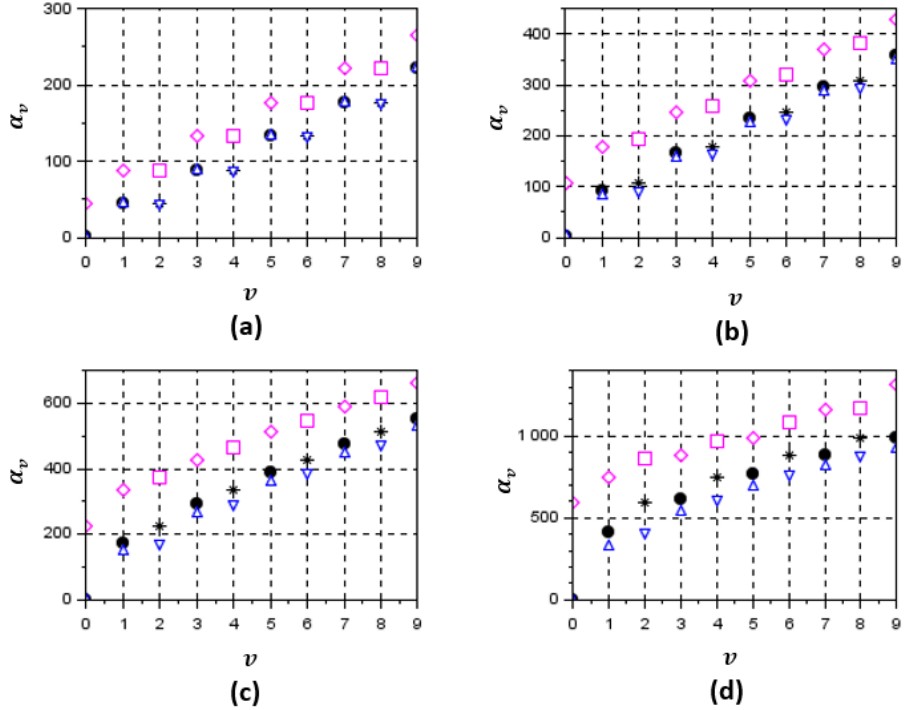

Figure 7: The variation of eigenvalue with the eigen-index for $m = 0$ with $-\sigma.B$ (black solid circles) and $+\sigma.B$ (black asterisks); $m = +1$ with $-\sigma.B$ (blue upward triangles) and $+\sigma.B$ (blue downward triangles); $m = -1$ with $-\sigma.B$ (magenta diamonds) and $+\sigma.B$ (magenta squares) for (a) $n = 0$, (b) $n = -0.3$, (c) $n = -0.5$, and (d) $n = -0.7$. Here, $B_0 = 10^{15}$ G pm$^{-n}$.

the net dispersion relation for $m = 0$ and $n \geq -0.5$ is

$$\alpha_\nu = C_3 \, B_0^{\frac{2}{n+2}} \, (\nu + C_5)^{\frac{2+2n}{n+2}} \left[ 1 \pm \frac{C_5}{(\nu + C_5)} \right]. \tag{36}$$

### 3.2.2 $m \neq 0$

When we probe the eigenvalues taking non-zero $m$, they show enthralling trends. We compare the eigenvalues between $m = 1$ and $m = -1$ along with $m = 0$ for $n = -0.3, -0.5$ and $-0.7$. As understood from Eq. (22) for the case of constant field and discussed below Eq. (22), positive $m$ does not have any impact on eigenvalues with respect to $m = 0$. Figure 7 shows how with decreasing $n$ (with more non-uniform field) eigenvalues for $m = 0$ and $m = 1$ along with $m = -1$ become distinct at a given eigen-index. It is clearly seen that as $n$ decreases, the difference between the eigenvalues for $m = 0$ and $m = 1$ increases.

Above behavior may be inferred to be an effect stemming from the rotation of particles and field behavior. Generally, rotation in the direction to the magnetic field is easier and to the opposite direction difficult. The case with of $m \neq 0$ implies that the electron has some angular momentum, which further implies its rotational motion. A positive $m$ means rotation in the direction to the magnetic field and negative implies opposite. When the magnetic field is homogeneous, a rotation in the direction to the magnetic field is not expected to require any extra force when there is no change of field magnitude (no force due to magnetic pressure), hence no change in energy. However, to rotate a particle opposite to its natural direction,

extra force is required, hence raising of energy. For an inhomogeneous field, as the magnitude of field changes (here decreases with distance), the particle has to overcome the force due to magnetic pressure, even if the direction is same as of the magnetic field. Hence, its own energy dissipates, leading to the less energy to align with the magnetic field.

# 4 Modification to thermodynamic properties and equation of state of cold degenerate electron gas

The main impact of the variation of LQ in the presence of varying magnetic field established above is to the systems where field varies drastically over the spatial scale. Since there is a huge difference between central and surface magnetic fields in the astrophysical bodies like white dwarfs and neutron stars, their realistic properties should be determined in the presence of variable magnetic fields, in place of an approximate constant field. However, the quantum effect is important, and the application of preceding discussion works out in stellar astrophysics, only if the variation of strong fields is in the length scale of gyromagnetic radius. Otherwise, even if LQ might play an important role to determine the underlying EoS and stellar structure depending on the field strength, uniform magnetic field based results suffice. In high densities, matter in white dwarfs and neutron stars turns out to be degenerate and their EoSs play indispensable role to determine the underlying stellar properties. In the presence of strong magnetic field, such a highly dense matter may get influenced by LQ, depending on the field strength, composition and density. The variation of magnetic field can be chosen appropriately by considering suitable $n$ in our chosen field ansatz, once central and surface magnetic fields are known or at best anticipated.

As emphasized above, modified LQ, based on non-uniform magnetic field, is useful in determining its effect in EoS only in pm scale if the magnetic field is of the order of $10^{15}$ G. The main effect of magnetic field via LQ is to modify the available density of states for electrons. For a constant field, the difference in energy levels is constant, but for a variable magnetic field case, the energy difference between levels no longer remains constant, as shown in Figure 4. Also there are separate sets of energy levels for spin-up and spin-down electrons. In the presence of variable magnetic field EoS can be found out as follows.

Considering only one kind of electrons at a time, say spin-down, the number of states per unit volume in a momentum interval $\Delta p_z$ for a Landau level $\nu$ for non-uniform energy levels is given by (generalized from [10])

$$\frac{\pi}{h^3}\left(\tilde{P}_{\nu+1}-\tilde{P}_\nu\right)\Delta p_z. \tag{37}$$

For a constant magnetic field and all electrons, the above expression is amended with a degeneracy factor $g_\nu$ of Landau levels, where $g_\nu = 1$ for the ground state and $g_\nu = 2$ for other states. However the situation is different for a non-uniform field.

Let us define $\left(\tilde{P}_{\nu+1}-\tilde{P}_\nu\right)_\pm = D(\nu)_\pm$. Therefore, the electron density of states in the absence of magnetic field $\frac{2}{h^3}\int d^3p$ is replaced by

$$\frac{2\pi}{h^3}D(\nu)_\pm\int dp_z, \tag{38}$$

in the case of a non-zero magnetic field.

In order to calculate the electron number density $n_e$ at zero temperature, we have to evaluate the integral in Eq. (38) from $p_z = 0$ to $p_F(\nu)$, which is the Fermi momentum of the Landau level $\nu$, and obtain

$$n_{e\pm} = \sum_{\nu=0}^{\nu_m}\frac{2\pi}{h^3}D(\nu)_\pm p_F(\nu). \tag{39}$$

The Fermi energy $E_F$ of the electrons for the Landau level $\nu$ is given by

$$E_F^2 = m_e^2 c^4 + p_F(\nu)^2 c^2 + \tilde{P}(\nu) c^2. \tag{40}$$

The upper limit $\nu_m$ of the summation, corresponding to the upper limit of levels, in Eq. (39) is derived from the condition that $p_F^2(\nu) \geq 0$, which implies

$$\tilde{P}(\nu) c^2 \leq E_F^2 - m_e^2 c^4 \tag{41}$$

or

$$\alpha_{\nu_m} = \epsilon_{F\,max}^2 - 1. \tag{42}$$

Hence, total electron density taking into account both the spins of electron is

$$n_e = n_{e+} + n_{e-} = \frac{1}{(2\pi)^2 \lambda_e^3} \left( \sum_{\nu=0}^{\nu=\nu_{m-}} \beta_-(\nu) x_{F-}(\nu) + \sum_{\nu=0}^{\nu=\nu_{m+}} \beta_+(\nu) x_{F+}(\nu) \right), \tag{43}$$

where $+$ sign indicates spin-up and $-$ sign spin-down, $x_F = p_F/m_e c$,

$$x_{F\pm}(\nu) = \left[ \epsilon_F^2 - (1 + \alpha_\pm(\nu)) \right]^{\frac{1}{2}} \tag{44}$$

and

$$\beta_\pm = (\alpha_\pm(\nu+1) - \alpha_\pm(\nu-1))/2. \tag{45}$$

The electron energy density at zero temperature is

$$
\begin{aligned}
\varepsilon_e &= \frac{1}{(2\pi)^2 \lambda_e^3} \left( \sum_{\nu=0}^{\nu=\nu_{m-}} \beta_-(\nu) \int_0^{x_{F-}(\nu)} E_{\nu,p_z} dx_z + \sum_{\nu=0}^{\nu=\nu_{m+}} \beta_+(\nu) \int_0^{x_{F+}(\nu)} E_{\nu,p_z} dx_z \right) \\
&= \frac{m_e c^2}{(2\pi)^2 \lambda_e^3} \left( \sum_{\nu=0}^{\nu=\nu_{m-}} \beta_-(\nu)(1 + \alpha_-(\nu)) f_1 \left[ \frac{x_{F-}(\nu)}{(1 + \alpha_-(\nu))^{1/2}} \right] \right. \\
&\quad \left. + \sum_{\nu=0}^{\nu=\nu_{m+}} \beta_+(\nu)(1 + \alpha_+(\nu)) f_1 \left[ \frac{x_{F+}(\nu)}{(1 + \alpha_+(\nu))^{1/2}} \right] \right),
\end{aligned}
\tag{46}
$$

where

$$f_1(z) = \frac{1}{2} \left( z \sqrt{1+z^2} + \ln(z + \sqrt{1+z^2}) \right) \tag{47}$$

and $E_{\nu,p_z}$ is the quantized energy levels defined in, e.g., Eq. (15). The pressure of an electron gas is given by

$$
\begin{aligned}
P_e &= n_e^2 \frac{d}{dn_e} \left( \frac{\varepsilon_e}{n_e} \right) = -\varepsilon_e + n_e E_F \\
&= \frac{m_e c^2}{(2\pi)^2 \lambda_e^3} \left( \sum_{\nu=0}^{\nu=\nu_{m-}} \beta_-(\nu)(1 + \alpha_-(\nu)) f_2 \left[ \frac{x_{F-}(\nu)}{(1 + \alpha_-(\nu))^{1/2}} \right] \right. \\
&\quad \left. + \sum_{\nu=0}^{\nu=\nu_{m+}} \beta_+(\nu)(1 + \alpha_+(\nu)) f_2 \left[ \frac{x_{F+}(\nu)}{(1 + \alpha_+(\nu))^{1/2}} \right] \right),
\end{aligned}
\tag{48}
$$

where

$$f_2(z) = \frac{1}{2} \left( z \sqrt{1+z^2} - \ln(z + \sqrt{1+z^2}) \right). \tag{49}$$

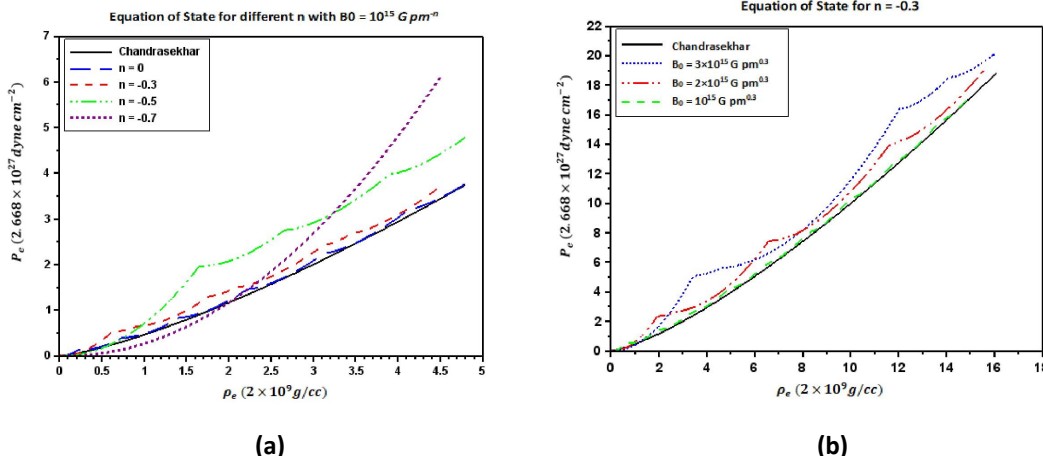

**(a)**          **(b)**

Figure 8: (a) Equation of state for $n = 0$ with $\epsilon_{Fmax} = 17$ (long-dashed blue line), $n = -0.3$ with $\epsilon_{Fmax} = 17$ (dashed red line), $n = -0.5$ with $\epsilon_{Fmax} = 18$ (dot-dashed green line) and $n = -0.7$ with $\epsilon_{Fmax} = 15$ (dotted violate line), for $B_0 = 10^{15}$ G pm$^{-n}$, along with Chandrasekhar's result with $\epsilon_{Fmax} = 17$ (solid black line), (b) Equation of state for $B_0 = 10^{15}$ G pm$^{-n}$ (dashed green line), $2 \times 10^{15}$ G pm$^{-n}$ (dot-dashed red line) and $3 \times 10^{15}$ G pm$^{-n}$ (dotted blue line) along with Chandrasekhar's result (solid black line) for $n = -0.3$ with $\epsilon_{Fmax} = 25$, when pressure is in units of $2.668 \times 10^{27}$ erg cm$^{-3}$ and mass density in units of $2 \times 10^9$ gm cm$^{-3}$.

We know that with the change in allowed number of levels in a system for a given $\epsilon_{Fmax}$, EoS changes significantly. As the number of level increases, the pressure decreases and increases for a given density, which are respectively called softer and harder/stiffer EoS, at a high and low densities respectively (see [10] for the example of constant field). Figure 8(a) shows EoS for various $n$. With the decrease in $n$, EoS becomes stiffer at a high density and softer at a low density, indicating lesser number of allowed levels in the system for low $n$. Figure 8(b) shows how EoS becomes stiffer, at the high density regime, with increasing $B_0$ for a fixed $n$. This is as per the expectation as stronger field leads to the more LQ effect with less number of levels populated, deviating the results more from the nonmagnetic case.

Note importantly that above EoSs shown in Figure 8 are applicable to a B-WD till the radius from the center where field does not decay significantly and hence LQ is still valid. On the other hand, depending on $n$, field decays with the radial coordinate in the pm scale and hence a given EoS does not remain valid for a B-WD beyond a scale of the order of pm.

# 5 Implications

## 5.1 Mass–Radius Relation of Magnetized White Dwarfs

An immediate astrophysical implication of LQ is to the mass–radius relation of (highly) magnetized white dwarfs. As mentioned in the Introduction (e.g., [11, 24]), strong magnetic field can significantly modify the mass-radius relation due to LQ as well as classical Lorentz force effects. However, it has been argued sometime [31] that LQ effect is not important in control-

ling stellar structure of white dwarfs and only Lorentz force would suffice the same. Here we plan to check if LQ has any impact on the white dwarf stellar structure.

For the present purpose, we consider a sample field profile in cylindrical polar coordinates as

$$
\begin{aligned}
\mathbf{B} &= B_0 \hat{z}, & &\text{for } \rho < 850 \text{ km}, \\
\mathbf{B} &= B_0 \left( \frac{\rho}{1 \text{ km}} \right)^{-0.37} \hat{z}, & &\text{for } \rho < 900 \text{ km}, \\
\mathbf{B} &= B_0 \left( \frac{\rho}{1 \text{ km}} \right)^{-0.99} \hat{z}, & &\text{otherwise},
\end{aligned}
\tag{50}
$$

so that $(\mathbf{B} \cdot \boldsymbol{\nabla})\mathbf{B} = 0$ and $\boldsymbol{\nabla} \cdot \mathbf{B} = 0$. Therefore, the nonrotating white dwarfs will be spherical in shape. Hence, in spherical polar coordinates with $\theta = \pi/2$, the field profile is given by

$$
\begin{aligned}
\mathbf{B} &= -B_0 \hat{\theta}, & &\text{for } r < 850 \text{ km}, \\
\mathbf{B} &= -B_0 \left( \frac{r}{1 \text{ km}} \right)^{-0.37} \hat{\theta}, & &\text{for } r < 900 \text{ km}, \\
\mathbf{B} &= -B_0 \left( \frac{r}{1 \text{ km}} \right)^{-0.99} \hat{\theta}, & &\text{otherwise}.
\end{aligned}
\tag{51}
$$

This profile assures (based on the solution for the stellar structure given below) that at the surface the field is restricted to be around $10^{12}$ G when $B_0 = 2 \times 10^{15}$ G.

Therefore, the mass and radius of a white dwarf can be obtained by solving

$$
\frac{d}{dr} \left( P_e + \frac{B^2}{8\pi} \right) = -\frac{GM(r)(\rho_e + \rho_B)}{r^2},
\tag{52}
$$

$$
\frac{dM(r)}{dr} = 4\pi r^2 (\rho_e + \rho_B),
\tag{53}
$$

where $\rho_B$ is the magnetic density, $B^2 = \mathbf{B} \cdot \mathbf{B}$, $\rho_e = n_e m_p \mu_e$, $m_p$ is the mass of proton, $\mu_e$ is the mean molecular weight per electron and $G$ is Newton's gravitation constant. Here for $r < 850$ km, EoS would be Landau quantized for $B_0 = 2 \times 10^{15}$ G, but for a uniform magnetic field. For $r \geq 850$ km, the field decays to a lower strength so that Chandrasekhar's nonmagnetic EoS suffices. Only at the interface around 850 km, non-uniform field based LQ applies in EoS, but in a very tiny region. Hence, for the present example, LQ based on uniform field practically influences EoS.

Figure 9(a) shows EoS for a constant magnetic field describing B-WD from centre to $r < 850$ km, along with Chandrasekhar's nonmagnetic EoS which is applicable in $r \geq 850$ km for the star with profile given by Eq. (51). Figures 9(b) and 10 show that the mass turns out to be significantly super-Chandrasekhar for the present field profile and the mass-limit may arise from the upper limit of density, e.g., arisen due to pycnonuclear reactions, neutron drip etc. We plan to explore this in detail in a future work, particularly the deviation from the mass-radius trend of Chandrasekhar with increasing $B_0$. Note that in principle with decreasing $\rho_c$, maximum field in a B-WD, i.e. $B_0$, should be decreasing. However, for uniformity and the convenience of comparison and computation, we have kept $B_0$ same for all the stars in the sample computations. This shows apparent decreasing mass with increasing $\rho_c$.

In Figure 10 we assess how important the LQ effect over the Lorentz force is, at least for the chosen profile. We find that a hypothetical case with Lorentz force but without LQ, i.e. with Chandrasekhar's EoS, leads the mass to restrict below the Chandrasekhar-limit. This is

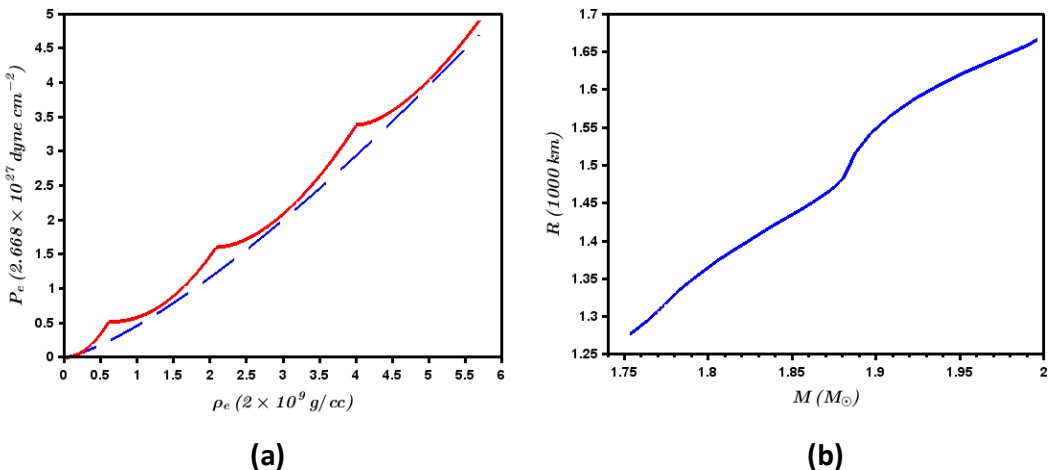

**(a)**                      **(b)**

Figure 9: (a) EoS for constant magnetic field ($n = 0$) of $B_{cent} = 2 \times 10^{15}$ G (red solid line) describing the central region of white dwarfs (see Eq. 51) and Chandrasekhar's EoS (dashed blue line) describing otherwise for $\epsilon_{Fmax} = 18$, (b) corresponding mass–radius relation.

understood as the high density B-WDs have smaller radius and according to the chosen profile given by Eq. (51) field remains constant throughout or almost throughout, hence there is no (significant) Lorentz force. Therefore, the mass is restricted mostly based on Chandrasekhar's EoS. However, lower density B-WDs, having radius larger than 850 km with the field varying in the outer region, exhibit Lorentz force, which is however not adequate enough to increase mass. Hence, LQ plays a significant role to bring in super-Chandrasekhar white dwarfs. Note that the hypothetical mass–radius relation exhibits two discrete branches (one being around $1.15 - 1.2 M_{\odot}$). This break of continuity in the mass is due to the sharp change in field due to its transition from constant to varying trends, which however does not arise in the realistic case with LQ effect included, as shown in Figure 9(b).

Of course, the chosen field profile is just a test sample, in particular to facilitate the exploration of LQ, keeping other physics intact. This however establishes that for a realistic case, the LQ effect should not be neglected at high magnetic fields.

## 5.2 Quantum Speed Limit

Quantum speed of particle determines how fast it transits from one energy level to another. It has a direct influence on the processing speed of quantum information. It was shown by Villamizar and Duzzoini [9] that for an electron in ground state with the up spin, the maximum quantum speed, also known as *quantum speed limit*, irrespective of the magnitude of magnetic field is $0.2407c$, if the magnetic field is uniform. We apply the same idea in the regime of non-uniform magnetic field.

The wavefunction of an electron with the up spin in state $\nu$ is given by

$$\Psi = e^{\frac{-iE_{\nu}t}{\hbar}} \psi_{\nu}, \tag{54}$$

such that

$$\psi_{\nu} = e^{i\left(m\phi + \frac{p_z}{\hbar}z\right)} \begin{bmatrix} R_{\nu+}(\rho) \\ -R_{\nu+}(\rho) \end{bmatrix}. \tag{55}$$

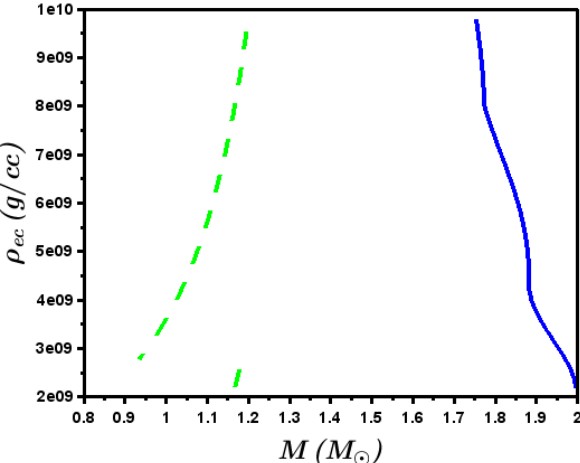

Figure 10: Comparison of $M - \rho_{ec}$ relation (solid blue line) with that of the hypothetical case with Chandrasekhar's EoS but Lorentz force intact (dashed green line).

Let the initial state of the spin-up electron be the superposition of two consecutive states, ground and first excited states, with $m = 0$ and $p_z = 0$, given by

$$\Psi(\rho, 0) = \frac{1}{\sqrt{2}} [\psi_0(\rho) + \psi_1(\rho)], \tag{56}$$

where the respective energies are $E_0$ and $E_1$.

Consider the evolution of wavefunction from ground state to first excited state. The minimum time of evolution is given by the Mandelstam-Tamm (MT) bound [32]

$$T_{min} = \frac{\pi \hbar}{2 \Delta H}, \tag{57}$$

where

$$\Delta H = \frac{E_1 - E_0}{2} \tag{58}$$

in this case.

The radial displacement of particle in time $T_{min}$ is

$$\rho_{disp} = |\langle \rho \rangle_{T_{min}} - \langle \rho \rangle_0| = 2 \left| \int_0^\infty \rho D_S(\rho) d\rho \right|, \tag{59}$$

where

$$D_s(\rho) = \psi_0^+ \, \rho \, \psi_1. \tag{60}$$

Thus, quantum speed of electron is given by

$$\tilde{v} = \frac{\rho_{disp}}{T_{min}}. \tag{61}$$

In order to determine quantum speed limit, we choose large $B_0$ ($= 10^{16}$ G pm$^{-n}$) such that on changing $B_0$ further, there is not much change in quantum speed for a given $n$.

As it can be seen from Figure 11, quantum speed of electron increases with increasing $n$, reaches maximum and then begins to decrease. The quantum speed limit increases compared
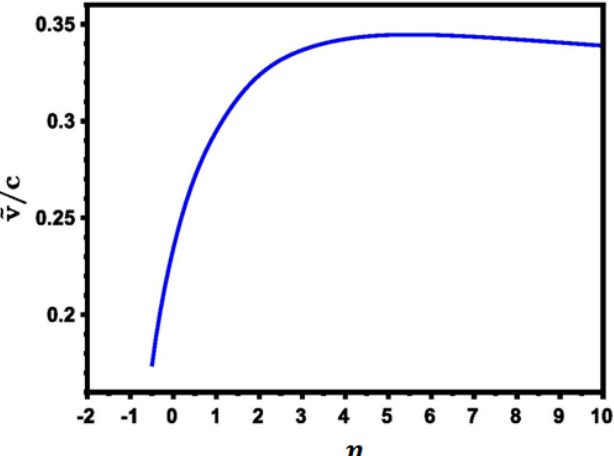

Figure 11: Variation of quantum speed of spin-up electron for transition from ground state to first excited state with different $n$ at $B_0 = 10^{16}$ G pm$^{-n}$.

to its value in a uniform magnetic field ($n = 0$) for $n > 0$. This is related to different rearrangements of energy levels lifting degeneracy between the fields with $n < 0$ and $n > 0$, shown in Figure 12. Thus, if we could trap an electron in a magnetic field which is spatially increasing in magnitude even linearly ($n = 1$) within a small scale, then we can achieve a higher speed of transition of electron. This can be extremely useful in faster processing of quantum information in the presence of variable magnetic field as compared to the uniform field. We plan to investigate this application of variable magnetic field in detail in a future work.

# 6  Conclusion

The LQ and the underlying dispersion relation with varying magnetic fields is a unique problem on its own. When the field is constant, the problem is nothing but a harmonic oscillator, whose analytical solution for energy is well-known. However for a varying field, the situation is quite different and difficult. Unless the magnetic field or more precisely the underlying vector potential follows a specific profile, e.g. a power-law variation, even a semi-analytical solution seems to be very difficult. We have chosen a simple power-law variation of the magnetic field and the corresponding vector potential, so that its satisfies no magnetic monopole condition and also magnetic tension vanishes. The latter helps applying this result in stellar physics easily.

For the ease of comparing with the constant field case, we develop the underlying quantum mechanics in the cylindrical coordinate system, which however can easily be recast for spherical coordinates. We have obtained a very important result. Due to the variation of magnetic field, the degeneracy in energy levels, as known for the constant field, is lifted and there is unique alignment of levels of spin-up and spin-down electrons depending on the nature of change of magnetic field. The result is not difficult to understand. As the field magnitude changes at each point, the LQ effect keeps varying at each point as well, which leads to non-overlapping energy levels as they are for constant field, hence lifting the degeneracy.

The non-uniform magnetic field has a wide range of applications in the both decaying as well as in rising regimes. If we consider the decaying magnetic field profile, the above result importantly has a significant consequence to the EoS of the magnetized degenerate electron

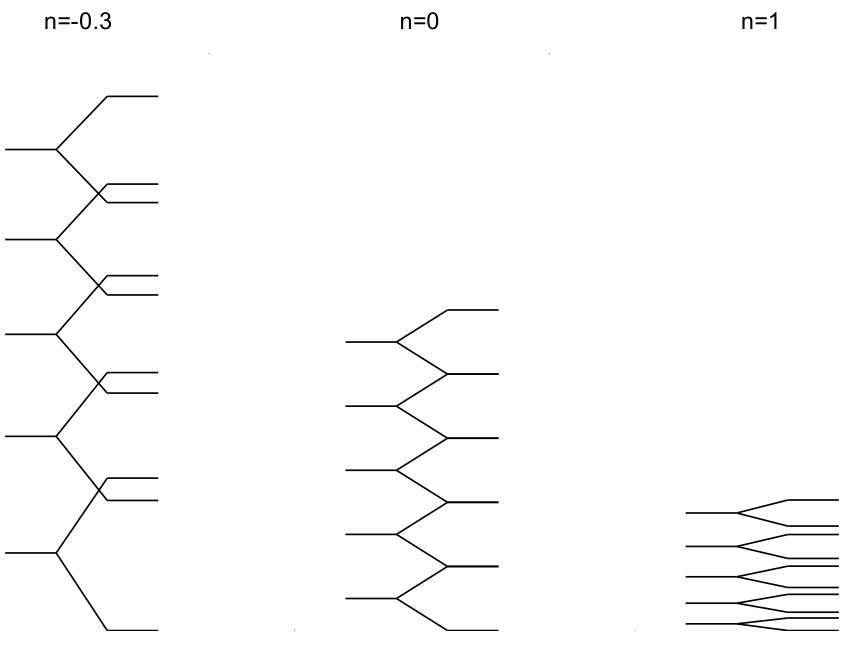

Figure 12: Comparison of schematic representation of the energy level splitting between the magnetic field with $n < 0$ and that with $n > 0$, along with a case of uniform magnetic field ($n = 0$).

gas. For a similar Fermi energy, at the low density regime, pressure decreases compared to the constant field case, while it is opposite in the high density regime, for a given density.

Modification to the EoS due to LQ further leads to the increase of white dwarf mass significantly compared to the mass without field. While an effect of constant field on to the white dwarf mass arises solely due to the LQ, the varying field also add to the Lorentz force. However, depending on the nature of field variation, the LQ effect plays an indispensable role to determine the stellar structure and the mass of white dwarfs. Our sample field profile chosen for a test case is not far from reality. Thus, it establishes that in a realistic situation, the LQ in a white dwarf with a strong magnetic field is an important effect. On the other hand, the spatially growing magnetic field could be proven useful in quantum information where we can achieve higher quantum speed of particle in the presence of variable magnetic field.

In general, LQ effects in variable magnetic field can be applied to a variety of physical systems ranging from astrophysics to quantum information to high energy physics to condensed matter. Suitable modifications to the effects such as quantum Hall effect in the presence of non-uniform magnetic field could give rise to unexplored but interesting experimental consequences.

## Acknowledgments

The authors thank Subhashish Banerjee of IIT Jodhpur and Diptiman Sen of IISc for reading the preliminary draft, discussion and suggestions. Also thanks are due to Surajit Kalita of IISc for cross-checking some of the mass–radius relations and discussion, and Debabrata Deb of IISc for useful comments on central magnetic field. S.A. thanks her undergraduate teacher S.

N. Sandhya of Miranda House, University of Delhi, for teaching the coding techniques used in this work.

## Author Contributions

B.M. proposed the problem and supervised the project; S.A. set up the numerical model and solved it; B.M. and S.A. formulated the problem, analyzed the numerical results and wrote the manuscript. G.G. verified the results and helped in improving the manuscript by adding physical subtleties to the work.

## A  Obtaining Dirac and Maxwell's equations from Lagrangian

The total Lagrangian density of the system of electrons of wave-function $\psi$ in the presence of electro-magnetic field is

$$\mathcal{L} = \bar{\psi}\left(i\gamma^\mu D_\mu - m\right)\psi - \frac{1}{16\pi}F_{\mu\nu}F^{\mu\nu}, \tag{62}$$

where

$$D_\mu = \partial_\mu - ieA_\mu, \quad F_{\mu\nu} = \partial_\mu A_\nu - \partial_\nu A_\mu, \tag{63}$$

in the units $\hbar = c = 1$, where $\mu$ runs from 0 to 3. Using Lagrangian equations of motion

$$\partial_\nu\left(\frac{\partial\mathcal{L}}{\partial(\partial_\nu\bar{\psi})}\right) - \frac{\partial\mathcal{L}}{\partial\bar{\psi}} = 0, \tag{64}$$

we obtain

$$\left(i\gamma^\mu D_\mu - m\right)\psi = 0, \tag{65}$$

which is the Dirac equation. Further using

$$\partial_\nu\left(\frac{\partial\mathcal{L}}{\partial(\partial_\nu A_\mu)}\right) - \frac{\partial\mathcal{L}}{\partial A_\mu} = 0, \tag{66}$$

we obtain

$$\partial_\nu F^{\nu\mu} - 4\pi j^\mu = 0, \tag{67}$$

which is the inhomogeneous Maxwell's equation, where $j^\mu$ is the current density, given by

$$j^\mu = e\bar{\psi}\gamma^\mu\psi. \tag{68}$$

For the time-independent magnetic field with vanishing electric field, Eq. (67) reduces to

$$\boldsymbol{\nabla} \times \mathbf{B} = 4\pi\mathbf{J}. \tag{69}$$

For the present purpose of Landau quantization in the presence of varying magnetic fields, we have solved the Eq. (65) above and obtained eigenvalues. On the other hand, for stellar structure, we have considered Eq. (69) in order to introduce Lorentz force proportional to $\mathbf{J} \times \mathbf{B}$.

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
