# Peer review of "Relativistic Landau quantization in non-uniform magnetic field and its applications to white dwarfs and quantum information"

_SciPost Physics, doi:SciPost Phys. 11, 093 (2021)_

## Round 1 · Referee Report · Anonymous · 2021-7-15

Strengths
1. The manuscript contains new and Important findings.
2. Those can be applied cutting across different areas of physics.
Weaknesses
1. It is not evident from the manuscript how big is the effect of the Landau quantisation (LQ) of electrons in non-uniform magnetic fields on the structures of white dwarfs (WDs).
Report
Authors should quantify the impact of the LQ on the mass-radius relation in non-uniform magnetic fields compared with the previous results.
Author: Banibrata Mukhopadhyay on 2021-08-16 [id 1673]
(in reply to Report 1 on 2021-07-15)
AUTHORS:
First, we would like to thank the reviewer very much for carefully reading
our manuscript and assessing it with encouraging comments. Here we respond
their comments one by one mentioning changes made in the manuscript. All the
changes are kept in bold-face and Figures 9 and 10 are replaced by new ones.
REVIEWER:
This work addresses the Landau Quantization (LQ) of electrons in non-uniform
magnetic fields and its applications to problems cutting across different
disciplines of physics, for example equation of state (EoS) and mass-radius
(M-R) relationship of white dwarfs (WDs) as well as the quantum speed of
electrons which might be important for processing the speed of quantum information.
The non-uniform field has a simple power law variation as given by B = B_0 ρ^n ẑ
where n=0 represents the uniform magnetic field.
It is a well known fact that the zeroth Landau level is singly degenerate whereas
it is doubly degenerate for all other levels in a uniform magnetic field. Authors
in this manuscript demonstrate that the degeneracy of energy levels as found in
a uniform field, is lifted in non-uniform magnetic fields. They further note that
there is a definite alignment of energy levels of spin-up and spin-down electrons
as the magnitude of a non-uniform magnetic field changes at each point. These are
new findings. It is also observed that the quantum speed of electrons increases as
'n' grows.
The impact of the LQ of electrons in non-uniform magnetic fields is investigated
on the EoS and M-R relationship of magnetized WDs. A specific example of the EoS for
n=-0.3 is used to compute the M-R curve in Figs. 9. If one carefully compares Fig. 9(a)
with Fig. 8(a), it is observed that the EoS of Fig. 8(a) for n=-0.3 is distinctly
stiffer with respect to the case of the uniform field (n=0), but it is not the
case in Fig. 9(a). This needs an explanation.
AUTHORS:
In Figure 8(a), when we showed EoS for a uniform magnetic field, it is for n=0
throughout the star and field B=B_0. However, in Figure 9a, we showed two EoSs
for the same star: one is EoS of central region with constant magnetic field
where B is higher than B_0, according to the chosen profile in the previous
version, and another is EoS with varying magnetic field. See in Eq. (5.2),
how central field and B_0 are related in the central region of the star,
for the model considered in the previous version, with a varying magnetic
field other than central region. Hence this difference is between Figure 8(a)
and Figure 9(a). In the revised version, however, we have replaced these figures
with better ones for clarity with modified physics and changed the text
accordingly. Note the modifications in section 5.1 in the revised version,
part of which prompted due to reviewer's this question.
REVIEWER:
It is observed in Fig. 8(a) that less number of Landau levels is populated
when n=-0.7 resulting in a softer EoS at lower densities and a stiffer EoS
at higher densities. It would be worth demonstrating the effect of the
non-uniform magnetic field for n=-0.7 on the M-R relation in a new plot.
AUTHORS:
For n=-0.7, the Lorentz force may turn out to be too strong for a realistic
central density of the star to destabilize it. Hence, we do not show
an M-R relation for n=-0.7. However, in the revised version, we have modified
the field profile for the M-R relation to remove this confusion. Note,
generally, that the trend of deviation for EoSs in nonuniform magnetic fields
compared to that of the uniform field is same for different n. However, the
deviation is more prominent with decreasing n, hence you noticed it for n=-0.7.
REVIEWER:
In the introduction, authors mention another alternative approach of the
non-uniform magnetic field in Ref.[27] extensively used in the investigations
of structures of neutron stars and WDs. Authors should compare their current
findings on the M-R relation of WDs with their previous results following
the prescription of the magnetic field variation of Ref. [27] inside the WDs.
This would definitely indicate whether new findings related to WDs in this
manuscript are quantitatively significant or not.
AUTHORS:
The aim of this work is not to propose a field profile opposing a density
dependent profile proposed earlier in Ref. 27, which facilitates M-R relations
influenced by magnetic effects (classical and quantum). The main aim is to show
the Landau quantization in nonuniform magnetic fields and how the quantization
effect changes w.r.to the uniform magnetic field case. For that we have
proposed a field profile, satisfying Maxwell's equations, and solved
the Dirac equation (as opposed to the case of uniform field, when the
Dirac equation has to be solved for a quadratic potential). Subsequently,
as one of the immediate applications, we have attempted to explore the
results in (highly) magnetized white dwarfs. Otherwise, the density dependent
field profile, as proposed in Ref. 27, is quite an effective one, at least
as long as the star is spherical and we quite like it. However, the solution
of Dirac equation in varying magnetic field can not be proceeded with a
density dependent magnetic field profile, which merely talks about field
magnitude at various density points. Nevertheless, in principle, as
indicated in the manuscript, by changing parameters, various field profiles
can be obtained from the density dependent profile as well and we may get
equivalent results of the present one as long as its application is to
white dwarfs. Unless the reviewer is very particular about it, we prefer to
undertake this venture in a future work, in order to avoid deviating from
the main flow of the present manuscript, which is Landau quantization in
varying magnetic fields.
Nevertheless, we have revised the manuscript throughout for a better clarity.
Anonymous on 2021-07-19 [id 1584]
The authors investigate the two-dimensional motion of relativistic cold electrons in the presence of spatially varying magnetic fields. In this way they explore Landau quantization in non-uniform magnetic fields. They apply this to magnetized white dwarfs, with realistic non-uniform magnetic fields involved simultaneously, with Lorentz force and Landau quantization affecting the underlying degenerate electron gas, exhibiting a significant violation of the Chandrasekhar mass-limit.
The detailed analysis of the impact of non-uniform magnetic field is interesting. Its a
nice piece of work which I envisage could have repercussions into various fields, such as,
astrophysics, condensed matter systems and quantum information.
It would be nice if they could work out in some detail, the impact of the field profile on an application, for example, the magnetized white dwarfs.
Author: Banibrata Mukhopadhyay on 2021-08-16 [id 1674]
(in reply to Anonymous Comment on 2021-07-19 [id 1584])AUTHORS:
First, we would like to thank the reviewer very much for encouraging comments
and suggestions. Here we respond their comments one by one mentioning changes
made in the manuscript. All the changes are kept in bold-face and Figures 9
and 10 are replaced by new ones.
REVIEWER:
The authors investigate the two-dimensional motion of relativistic cold electrons
in the presence of spatially varying magnetic fields. In this way they explore
Landau quantization in non-uniform magnetic fields. They apply this to magnetized
white dwarfs, with realistic non-uniform magnetic fields involved simultaneously,
with Lorentz force and Landau quantization affecting the underlying degenerate electron
gas, exhibiting a significant violation of the Chandrasekhar mass-limit.
The detailed analysis of the impact of non-uniform magnetic field is interesting.
Its a nice piece of work which I envisage could have repercussions into various fields,
such as, astrophysics, condensed matter systems and quantum information.
It would be nice if they could work out in some detail, the impact of the field
profile on an application, for example, the magnetized white dwarfs.
AUTHORS:
In Figures 8 and 9, we had shown that how the field profile for nonuniform
magnetic fields leads to modified EoS and subsequently which may lead to
super-Chandrasekhar white dwarfs. Interestingly, we had/have shown in Figure 10
that had we considered only Lorentz force along with Chandrasekhar's EoS
(but not Landau quantization based EoS), then the mass of white dwarfs
decreased significantly. This proves that Landau quantization has a significant
impact on the result. Nevertheless, in the revised version, we have included
a revised M-R relations for clarity along with a modified field profile more
suitable for the present purpose of M-R relations. See the replaced
Figures 9 and 10 and changes in the text in section 5.1. For further detailed
applications of Landau quantization obtained here and more detailed M-R relations
of white dwarfs, quantum speed etc., we prefer to refrain us for future works.
This will help to keep the main flow of the present manuscript intact, which is
the modification of Landau quantization in varying magnetic fields.
Nevertheless, we have revised the manuscript throughout for a better clarity.

---

## Round 2 · Referee Report · Anonymous (Referee 1) · 2021-9-8

Strengths

New and important findings.

Weaknesses

Weakness as mentioned in the previous report is addressed satisfactorily in the revised version.

Report

Authors have responded to my comments/queries adequately and revised the manuscript accordingly.

Requested changes

NIL

---

## Round 2 · Author Response

RESPONSE TO THE REVIEWER 1:

AUTHORS:

First, we would like to thank the reviewer very much for carefully reading our manuscript and assessing it with encouraging comments. Here we respond their comments one by one mentioning changes made in the manuscript. All the changes are kept in bold-face and Figures 9 and 10 are replaced by new ones.

REVIEWER:

This work addresses the Landau Quantization (LQ) of electrons in non-uniform magnetic fields and its applications to problems cutting across different disciplines of physics, for example equation of state (EoS) and mass-radius (M-R) relationship of white dwarfs (WDs) as well as the quantum speed of electrons which might be important for processing the speed of quantum information. The non-uniform field has a simple power law variation as given by B = B_0 ρ^n ẑ where n=0 represents the uniform magnetic field.

It is a well known fact that the zeroth Landau level is singly degenerate whereas it is doubly degenerate for all other levels in a uniform magnetic field. Authors in this manuscript demonstrate that the degeneracy of energy levels as found in a uniform field, is lifted in non-uniform magnetic fields. They further note that there is a definite alignment of energy levels of spin-up and spin-down electrons as the magnitude of a non-uniform magnetic field changes at each point. These are new findings. It is also observed that the quantum speed of electrons increases as 'n' grows.

The impact of the LQ of electrons in non-uniform magnetic fields is investigated on the EoS and M-R relationship of magnetized WDs. A specific example of the EoS for n=-0.3 is used to compute the M-R curve in Figs. 9. If one carefully compares Fig. 9(a) with Fig. 8(a), it is observed that the EoS of Fig. 8(a) for n=-0.3 is distinctly stiffer with respect to the case of the uniform field (n=0), but it is not the case in Fig. 9(a). This needs an explanation.

AUTHORS:

In Figure 8(a), when we showed EoS for a uniform magnetic field, it is for n=0 throughout the star and field B=B_0. However, in Figure 9a, we showed two EoSs for the same star: one is EoS of central region with constant magnetic field where B is higher than B_0, according to the chosen profile in the previous version, and another is EoS with varying magnetic field. See in Eq. (5.2), how central field and B_0 are related in the central region of the star, for the model considered in the previous version, with a varying magnetic field other than central region. Hence this difference is between Figure 8(a) and Figure 9(a). In the revised version, however, we have replaced these figures with better ones for clarity with modified physics and changed the text accordingly. Note the modifications in section 5.1 in the revised version, part of which prompted due to reviewer's this question.

REVIEWER:

It is observed in Fig. 8(a) that less number of Landau levels is populated when n=-0.7 resulting in a softer EoS at lower densities and a stiffer EoS at higher densities. It would be worth demonstrating the effect of the non-uniform magnetic field for n=-0.7 on the M-R relation in a new plot.

AUTHORS:

For n=-0.7, the Lorentz force may turn out to be too strong for a realistic central density of the star to destabilize it. Hence, we do not show an M-R relation for n=-0.7. However, in the revised version, we have modified the field profile for the M-R relation to remove this confusion. Note, generally, that the trend of deviation for EoSs in nonuniform magnetic fields compared to that of the uniform field is same for different n. However, the deviation is more prominent with decreasing n, hence you noticed it for n=-0.7.

REVIEWER:

In the introduction, authors mention another alternative approach of the non-uniform magnetic field in Ref.[27] extensively used in the investigations of structures of neutron stars and WDs. Authors should compare their current findings on the M-R relation of WDs with their previous results following the prescription of the magnetic field variation of Ref. [27] inside the WDs. This would definitely indicate whether new findings related to WDs in this manuscript are quantitatively significant or not.

AUTHORS:

The aim of this work is not to propose a field profile opposing a density dependent profile proposed earlier in Ref. 27, which facilitates M-R relations influenced by magnetic effects (classical and quantum). The main aim is to show the Landau quantization in nonuniform magnetic fields and how the quantization effect changes w.r.to the uniform magnetic field case. For that we have proposed a field profile, satisfying Maxwell's equations, and solved the Dirac equation (as opposed to the case of uniform field, when the Dirac equation has to be solved for a quadratic potential). Subsequently, as one of the immediate applications, we have attempted to explore the results in (highly) magnetized white dwarfs. Otherwise, the density dependent field profile, as proposed in Ref. 27, is quite an effective one, at least as long as the star is spherical and we quite like it. However, the solution of Dirac equation in varying magnetic field can not be proceeded with a density dependent magnetic field profile, which merely talks about field magnitude at various density points. Nevertheless, in principle, as indicated in the manuscript, by changing parameters, various field profiles can be obtained from the density dependent profile as well and we may get equivalent results of the present one as long as its application is to white dwarfs. Unless the reviewer is very particular about it, we prefer to undertake this venture in a future work, in order to avoid deviating from the main flow of the present manuscript, which is Landau quantization in varying magnetic fields.

Nevertheless, we have revised the manuscript throughout for a better clarity.

RESPONSE TO THE REVIEWER 2:

AUTHORS:

First, we would like to thank the reviewer very much for encouraging comments and suggestions. Here we respond their comments one by one mentioning changes made in the manuscript. All the changes are kept in bold-face and Figures 9 and 10 are replaced by new ones.

REVIEWER:

The authors investigate the two-dimensional motion of relativistic cold electrons in the presence of spatially varying magnetic fields. In this way they explore Landau quantization in non-uniform magnetic fields. They apply this to magnetized white dwarfs, with realistic non-uniform magnetic fields involved simultaneously, with Lorentz force and Landau quantization affecting the underlying degenerate electron gas, exhibiting a significant violation of the Chandrasekhar mass-limit.

The detailed analysis of the impact of non-uniform magnetic field is interesting. Its a nice piece of work which I envisage could have repercussions into various fields, such as, astrophysics, condensed matter systems and quantum information.

It would be nice if they could work out in some detail, the impact of the field profile on an application, for example, the magnetized white dwarfs.

AUTHORS:

In Figures 8 and 9, we had shown that how the field profile for nonuniform magnetic fields leads to modified EoS and subsequently which may lead to super-Chandrasekhar white dwarfs. Interestingly, we had/have shown in Figure 10 that had we considered only Lorentz force along with Chandrasekhar's EoS (but not Landau quantization based EoS), then the mass of white dwarfs decreased significantly. This proves that Landau quantization has a significant impact on the result. Nevertheless, in the revised version, we have included a revised M-R relations for clarity along with a modified field profile more suitable for the present purpose of M-R relations. See the replaced Figures 9 and 10 and changes in the text in section 5.1. For further detailed applications of Landau quantization obtained here and more detailed M-R relations of white dwarfs, quantum speed etc., we prefer to refrain us for future works. This will help to keep the main flow of the present manuscript intact, which is the modification of Landau quantization in varying magnetic fields.

Nevertheless, we have revised the manuscript throughout for a better clarity.

---

## Round 2 · List of Changes

The changes are mentioned while responding to the reviewers.

---

## Editorial Decision

published